# Essential Functions of Calmodulin and Identification of Its Proximal Interacting Proteins in Tachyzoite-Stage *Toxoplasma gondii* via BioID Technology

Yongle Song,[a,b] Longjiao Li,[a,b] Xinyu Mo,[a,b] Ming Pan,[a,b] Bang Shen,[a,b] Rui Fang,[b] Min Hu,[a,b] Junlong Zhao,[a,b] Yanqin Zhou[a,b]

aKey Laboratory Preventive Veterinary of Hubei Province, College of Veterinary Medicine, Huazhong Agricultural University, Wuhan, Hubei, People's Republic of China
bState Key Laboratory of Agricultural Microbiology, Huazhong Agricultural University, Wuhan, Hubei, People's Republic of China

**ABSTRACT** *Toxoplasma gondii* (*T. gondii*) is a pathogen belonging to the apicomplexan phylum, and it threatens human and animal health. Calcium ions, a critical second messenger in cells, can regulate important biological processes, including parasite invasion and egress. Calmodulin (CaM) is a small, highly conserved, $Ca^{2+}$-binding protein found in all eukaryotic cells. After binding to $Ca^{2+}$, CaM can be activated to interact with various proteins. However, little is known about CaM's function and its interacting proteins in *T. gondii*. In this study, we successfully knocked down CaM in the *T. gondii* parent strain TATI using a tetracycline-off system with the *Toxoplasma* CaM promoter. The results indicated that CaM was required for tachyzoite proliferation, invasion, and egress and that CaM depletion resulted in apicoplast loss, thus threatening parasite survival in the next lytic cycle. In the tachyzoite stage, CaM loss caused significant anomalies in the parasite's basal constriction, motility, and parasite rosette-like arrangement in the parasitophorous vacuole (PV). These phenotypic defects caused by CaM depletion indicate the importance of CaM in *T. gondii*. Therefore, it is important to identify the CaM-interacting proteins in *T. gondii*. Applying BioID technology, more than 300 CaM's proximal interacting proteins were identified from *T. gondii*. These CaM partners were broadly distributed throughout the parasite. Furthermore, the protein interactome and transcriptome analyses indicated the potential role of CaM in ion binding, cation binding, metal ion binding, calcium ion binding, and oxidation-reduction. Our findings shed light on the CaM function and CaM-interactome in *T. gondii* and other eukaryotes.

**IMPORTANCE** *Toxoplasma gondii* is an intracellular pathogen that threatens human and animal health. This unicellular parasite is active in many biological processes, such as egress and invasion. The implementation efficiency of *T. gondii* biological processes is dependent on signal transmission. $Ca^{2+}$, as a second messenger, is essential for the parasite's life cycle. Calmodulin, a ubiquitous $Ca^{2+}$ receptor protein, is highly conserved and mediates numerous $Ca^{2+}$-dependent events in eukaryotes. Few CaM functions or regulated partners have been characterized in *T. gondii* tachyzoites. Here, we reported the essential functions of calmodulin in *T. gondii* tachyzoite and the identification of its interacting partners using BioID technology, shedding light on the CaM function and CaM-interactome in *Toxoplasma gondii* and other eukaryotes.

**KEYWORDS** *Toxoplasma gondii*, calmodulin, tetracycline-off, protein interaction, transcriptome

Address correspondence to Yanqin Zhou, yanqinzhou@mail.hzau.edu.cn.

The authors declare no conflict of interest.

Apicomplexan parasites cause a wide range of diseases in humans and animals. *Toxoplasma gondii*, a model organism of apicomplexan parasites, is a ubiquitous intracellular parasite that causes chronic infections in warm-blooded animals. *T. gondii* affects one-third of the world's population and causes acute toxoplasmosis in immunocompromised individuals and newborns with inadequate immunity (1). The pathogenicity

of *T. gondii* is primarily due to its efficient lytic cycle and dissemination in host cells, which is closely linked to the parasite's active adaptation to host immune status (2). Fast-growing tachyzoites will develop into slow-growing bradyzoites (cysts) in skeletal muscle and the brain when they are confronted with the host's strong immunity and unfavorable cellular microenvironment. The tissue cysts can exist in the host for a long time and reactivate into tachyzoites when immunity is weak (3, 4).

During *T. gondii* infection, many biological processes require fast signal transmission in response to changing environmental conditions. Calcium ions ($Ca^{2+}$) are universal signaling molecules that regulate numerous cellular pathways. $Ca^{2+}$ and their interaction network are involved in many biological functions in eukaryotes, including contraction, secretion, cell division and differentiation, and sodium and potassium permeability in eukaryotes (5, 6). $Ca^{2+}$ fluctuation in the cytosol of the parasite and host cell affects some essential processes in *T. gondii*, including secretion, adhesion, invasion, motility, and egress (7–11). Compared with toxoplasma tachyzoites, bradyzoites exhibit suppressed $Ca^{2+}$ signaling, thus limiting egress and rapid response to external stimuli (12). *T. gondii* can increase cytoplasmic $Ca^{2+}$ content by acquiring $Ca^{2+}$ from external and intracellular calcium reserves (including the endoplasmic reticulum, mitochondria, acidocalcisome, and plant-like vacuole) (6, 13). The EF-hand domain is a $Ca^{2+}$-binding motif consisting of a helix-loop-helix topology with roughly 12 amino acids and some conserved Asp residues (14). Many calcium-binding proteins have the EF-hand domain, which can be regulated by calcium signaling. *T. gondii* genome encodes 43 proteins with the EF-hand domain, including calmodulin, calmodulin-like protein, calcium-dependent protein kinases (CDPKs), centrins, serine/threonine protein phosphatases, and hypothetical proteins with uncertain functions.

CaM is a small calcium-modulated protein found in all eukaryotic cells. CaM can affect signaling pathways by regulating various crucial processes such as growth, proliferation, and movement. CaM is evolutionarily highly conserved and contains four $Ca^{2+}$-binding EF hands. When it binds to $Ca^{2+}$ reversibly, CaM can exhibit a range of conformations and interact with various target proteins, such as motor proteins, ion channels, kinases, phosphatases, and membrane transport proteins (15–17). CaM can interact with targets through CaM-binding motifs, broadly categorized into classical $Ca^{2+}$-dependent binding motifs and $Ca^{2+}$-independent binding motifs, such as the IQ and IQ-like motifs (18). In addition, CaM also interacts with some target proteins in the presence of $Ca^{2+}$ by regulating CaM conformation. These targets binding to the CaM with different conformations could not be easily identified through conventional motif search.

The *T. gondii* genome encodes many CaM-like proteins. About a dozen CaM-like proteins were initially identified in *T. gondii* by conserved ortholog search and phylogenetic comparison, including CaM, centrins, and CaM-like proteins (19). These CaM-like proteins were localized in the conoid (such as CaM1, 2, and 3), the centrosome (such as Centrin 1 and 2), and the plasma membrane (such as myosin light chain 2) (20–22). The partial functions of some CaM-like proteins and centrins in *T. gondii* have been characterized in detail by CRISPR-Cas9 editing technology. However, little work on the 16.8 kDa highly conserved CaM (ToxoDB gene ID TGGT1_249240) in *T. gondii* has been performed, and almost all knowledge of CaM comes from other model organisms. We only know that CaM can be modulated by $Ca^{2+}$, located at the apical and basal ring, and it may act as a regulator of calcineurin in *T. gondii* (23, 24). Therefore, it is crucial to reveal the role of CaM in *T. gondii* and identify CaM interacting proteins because these interacting proteins may be potential drug targets against toxoplasma and other apicomplexan parasites.

## RESULTS

**CaM knockdown using an improved tetracycline-off system.** Although *T. gondii* encodes many CaM-like proteins with features such as low molecular weight and multiple EF-hands, the canonical CaM ortholog remains resistant to direct depletion, as

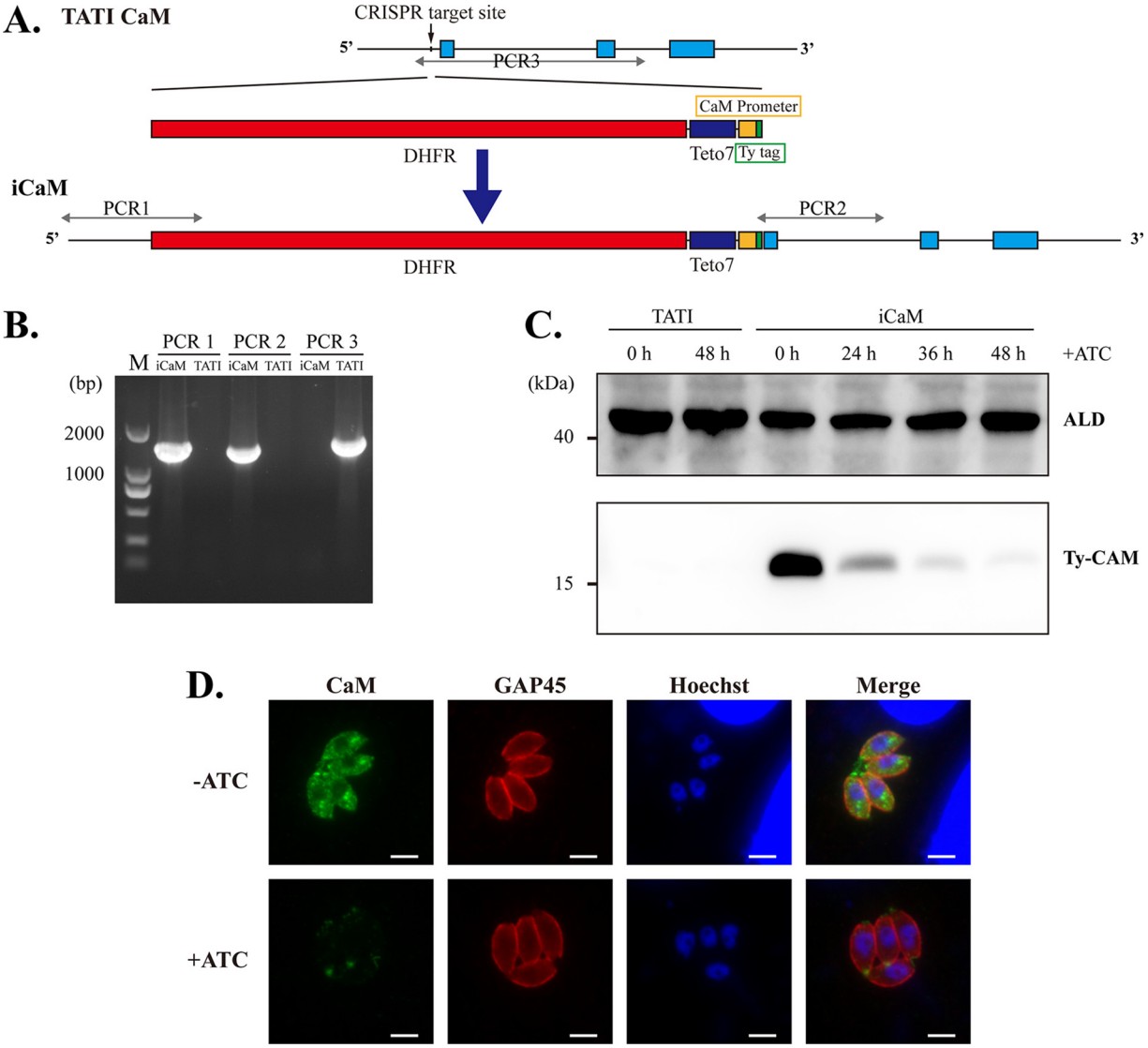

**FIG 1** Generation of the CaM knockdown system using an upgraded tetracycline-off system in *T. gondii*. (A) The construction pattern of the CaM knockdown system using CaM tetracycline-repressive promoter (CaM promoter) through CRISPR-Cas9 technology in the TATI parent strain. (B) Diagnostic PCRs on an iCaM clone. (C) Western blot of iCaM after 0, 24, 36, and 48 h of ATc treatment. TATI was used as the negative control. (D) IFA of iCaM pretreated with or without ATc for 24 h. Scale bar 5 $\mu$m.

demonstrated by our CRISPR-Cas9 experiment (Fig. S1A in Supplemental File 6). Consequently, the tetracycline-off system was employed in *T. gondii* to knock down CaM by replacing the endogenous promoter with a tetracycline-repressed promoter (25). None of our attempts to insert the common 7tetO-SAG1 minimal promoter element into the N terminus of CaM in the TATI parental strain were successful. Fortunately, 10 days after adding anhydrotetracycline hydrochloride (ATc), no visible plaque defects were found in two clones obtained after PCR diagnosis (Fig. S1B in Supplemental File 6). However, the ty-tag was undetectable in the clones with indirect immunofluorescence assay (IFA). To explore the possible reasons, the relevant elements were PCR amplified and sequenced. The sequencing results showed that the minimal SAG1 promoter and ty-tag at the CaM's N terminus reverted to CaM's initial state, and only a few tetO elements were present (Fig. S1C and D in Supplemental File 6). Therefore, we attempted to knock down CaM through the adjusted tetracycline-off system (iCaM, Fig. 1A) with the 7tetO-CaM minimal endogenous promoter. Diagnostic PCR (PCR1, 2, and 3) confirmed the correct integration of the iCaM in a single clone

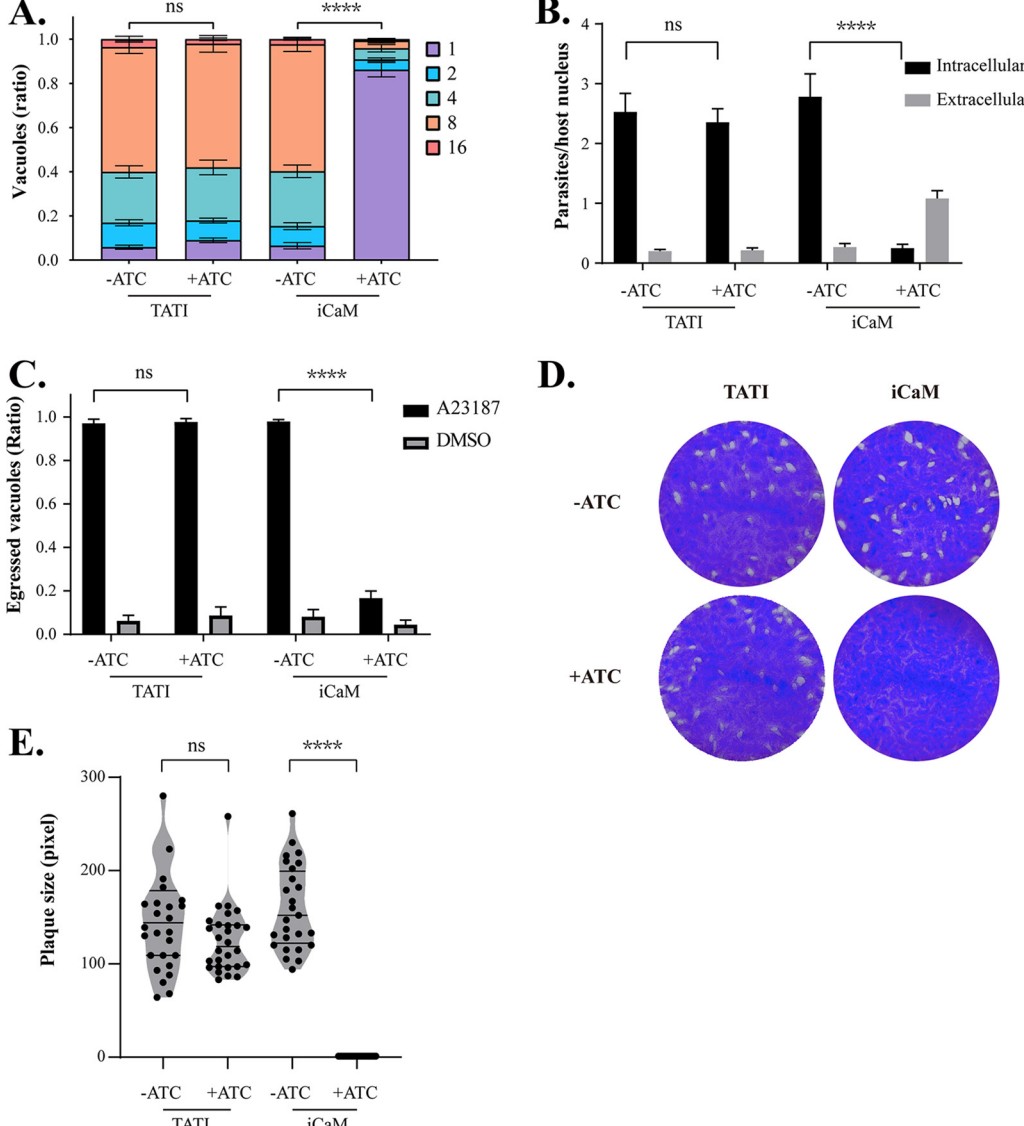

**FIG 2** Phenotypes induced by CaM knockdown in *T. gondii*. (A) The 24 h intracellular replication assay of iCaM was pretreated in the presence or absence of ATc for 48 h in HFF cells, as determined by the number of tachyzoites in each PV. Means ± SEM of three independent experiments. ****, $P < 0.0001$, two-way analysis of variance (ANOVA) with Tukey's multiple-comparison test. (B) HFF invasion assay of iCaM pretreated with or without ATc for 48 h. The mean number of parasites per host cell was calculated on each coverslip. Means ± SEM of three independent experiments. ****, $P < 0.0001$, one-way ANOVA with Tukey's multiple-comparison test. (C) Egress of iCaM pretreated with or without ATc for 36 h. The egressed vacuole ratio was calculated in each coverslip. Means ± SEM of three independent experiments. ****, $P < 0.0001$, one-way ANOVA with Tukey's multiple-comparison test. (D) Plaque assay of iCaM pretreated with or without ATc for 10 days. (E) The plaque size from (D). The experiment was repeated three times independently, each with three replicates. ****, $P < 0.0001$, unpaired Student's *t* test.

(Fig. 1B). Western blotting showed that CaM had almost disappeared with ATc for 36 h (Fig. 1C). IFA results indicated that the adjusted tetracycline-off system could be used to perform CaM knockdown. Additionally, CaM was found primarily in the apical, basal, and cytoplasm of intracellular tachyzoites (Fig. 1D).

**Knockdown of CaM caused various lethal phenotypes in tachyzoites.** The following experiments were conducted on an iCaM single clone using the adjusted tetracycline-off system. Significant defects in the replication, invasion, and egress phenotypes were detected in iCaM after ATc treatment, but the control TATI exhibited no similar phenotype defects (Fig. 2A to C). Compared to the iCaM without additional ATc and TATI with/without Atc control groups, the iCaM group formed no plaques with ATc for

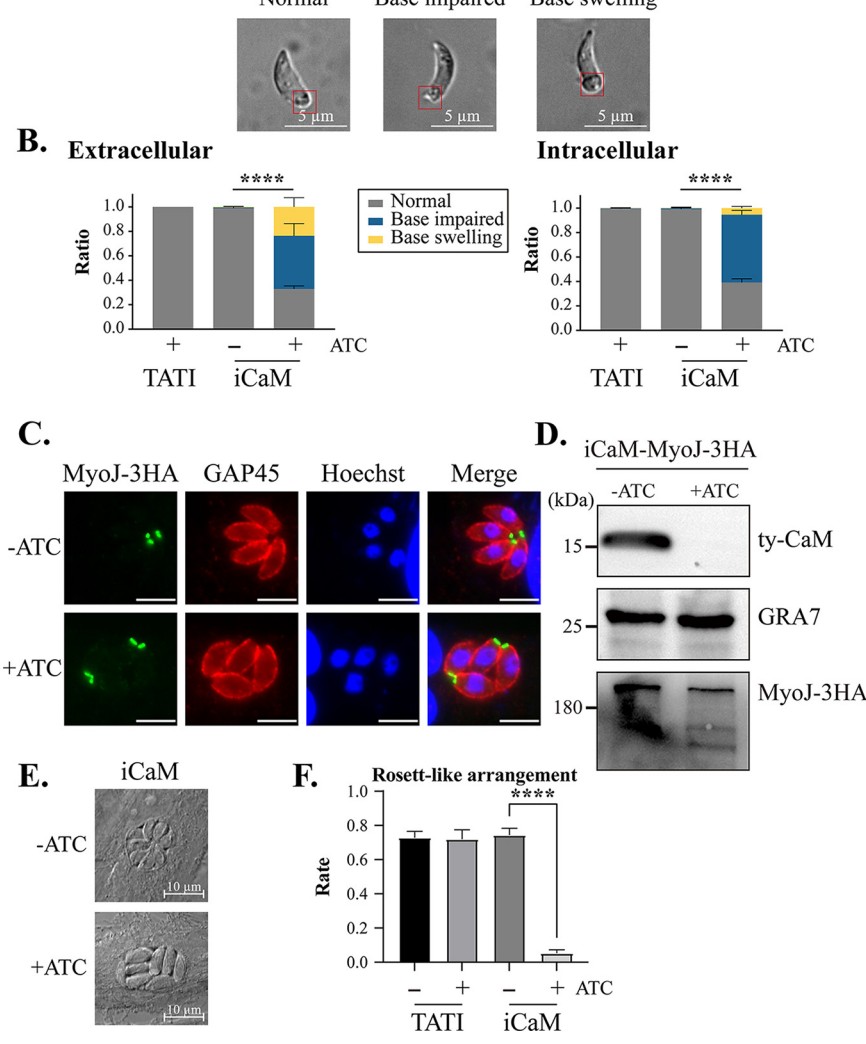

**FIG 3** Parasite base morphology and parasite arrangement in PV. (A) Base morphology of tachyzoites losing CaM. (B) The ratio of iCaM tachyzoite with base impaired, swelling, and normal in extracellular and intracellular, respectively. Means ± SD of three independent experiments. ****, $P < 0.0001$, two-way ANOVA with Tukey's multiple-comparison test. (C and D) IFA and Western blot of TgMyoJ tagged with 3HA in iCaM strain. (C) Scale bar 5 $\mu$m. (E) Arrangement of the parasite with or without CaM in PV. (F) Rosette-like arrangement of the parasite in PV. Three independent experiments. ****, $P < 0.0001$, Mean ± SD, unpaired Student's $t$ test.

10 days (Fig. 2D and E). Furthermore, there was no plaque formation when ATc was used for 5 days and then taken away for 5 days (Fig. S2 in Supplemental File 6). These results indicated that CaM was essential for tachyzoites and that ATc removal failed to rescue the defective phenotypes.

**CaM knockdown resulted in substantial abnormalities in the basal constriction and parasite motility.** After CaM was knocked down with ATc, aberrant posterior morphology of parasites inside and outside host cells was observed (Fig. 3A and B). TgMyoJ, a posterior pole protein, is in the base of tachyzoites and is implicated in the constriction of the basal complex (26). Therefore, TgMyoJ was chosen to identify the basal feature by inserting 3HA endogenous tags at the C terminus. CaM depletion resulted in the abnormal posterior pole in tachyzoites, which was visible using the TgMyoJ marker (Fig. 3C). The protein abundance of TgMyoJ was also slightly reduced but still detectable, revealed by Western blotting (Fig. 3D). These results showed that CaM depletion affected the parasite's posterior constriction.

We also observed that although ATc was only present for 24 h, the parasites

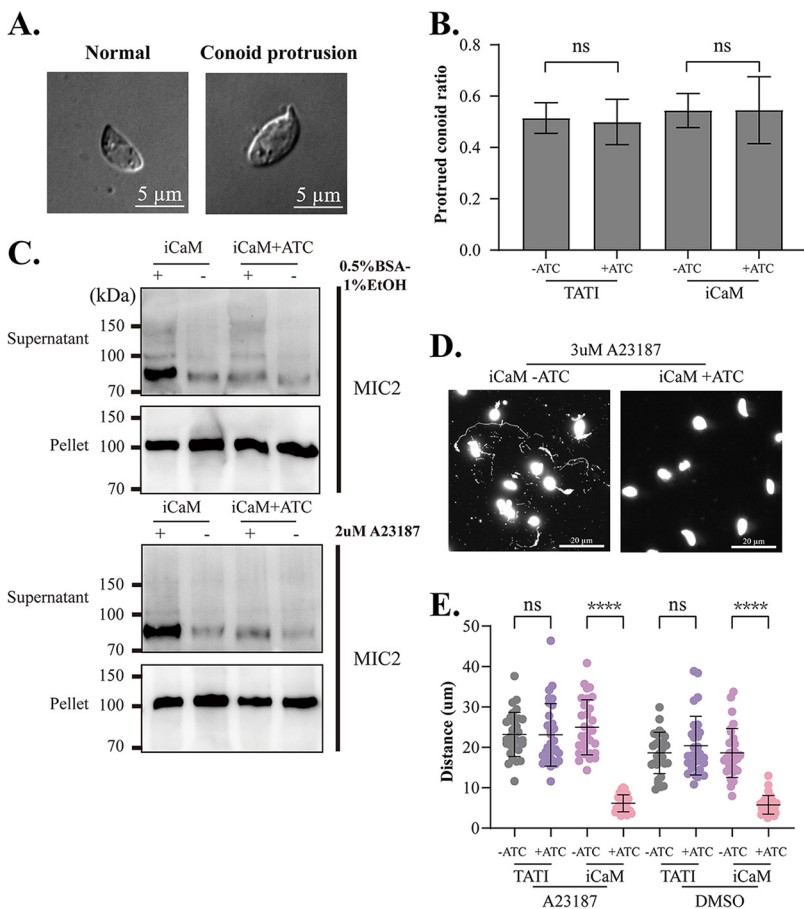

**FIG 4** Parasite defect in microneme secretion and motility upon CaM depletion. (A) Conoid protrusion stimulated with A23187. (B) Column graph of conoid protrusion stimulated with A23187. (C) Western blot of *T. gondii* microneme protein 2 (MIC2) secretion stimulated by BSA/EtOH and A23187, respectively. (D) IFA of parasite motility on coverslip stained with mouse anti-SAG1. The parasite was pretreated with or without ATc for 36 h, then put on a coverslip stimulated with A23187 for 20 min. (E) Scatterplot of parasite motility. (B, E) Means $\pm$ SD of three times independent experiments. ****, $P < 0.0001$, unpaired Student's *t* test.

without CaM rarely formed typical intracellular rosettes and were disorderly in their parasitophorous vacuoles (Fig. 3E and F). These findings implied that the parasites lacking CaM were incapable of maintaining the normal structure of the basal complex and the connection between tachyzoite and residual body (RB). RB, the structure derived from the mother tachyzoite, contributes to intravacuolar parasites' rosette-like arrangement and is the parental remnant after division (27).

The conoid, a distinctive structure at the parasite's extreme apex for *T. gondii* entry and escape from host cells, protruded after adding A23187 (28). The iCaM +ATc group had the same conoid protrusion as the control group induced by A23187 (Fig. 4A and B). However, Western blot analysis of MIC2 revealed that microneme secretion triggered by bovine serum albumin (BSA)-ethanol or A23187 in the iCaM +ATc group was considerably decreased compared with that in the control groups (Fig. 4C). Furthermore, the observation under the optical microscope showed that the tachyzoites with their CaM knocked down showed sluggish movement. Therefore, the motility of parasites was assessed by tracking the motion trajectory. The iCaM +ATc group exhibited a much shorter motion trajectory than the control group (Fig. 4D). Furthermore, raising the parasite cytoplasm $Ca^{2+}$ level failed to compensate for the motility defect of parasites lacking CaM (Fig. 4E).

**CaM depletion caused the loss of apicoplast.** Nearly all known members of the Apicomplexa have an apicoplast, a plastid-like organelle surrounded by four membranes,

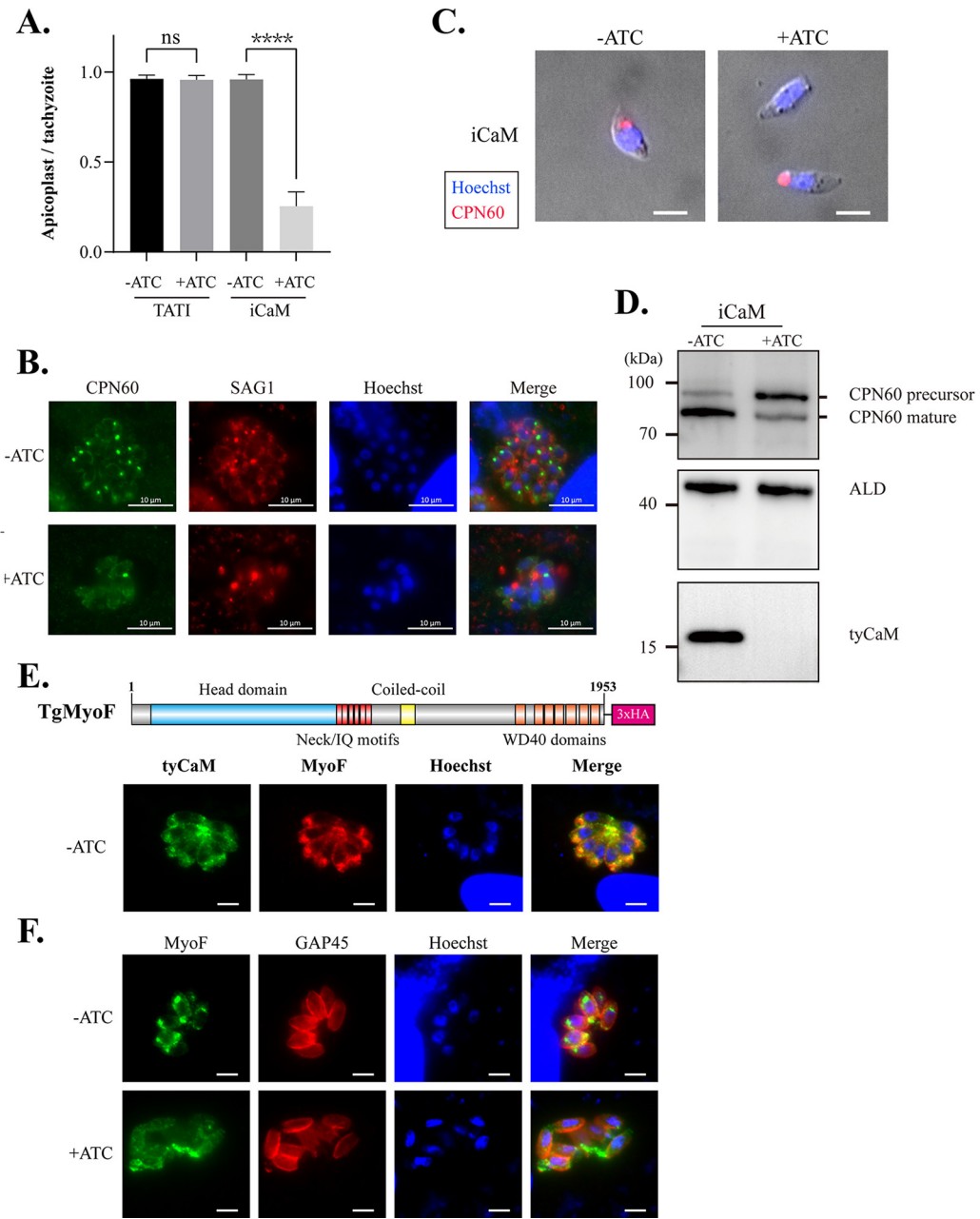

**FIG 5** CaM depletion-induced apicoplast loss. (A) Column graph of apicoplast per tachyzoite. Means $\pm$ SD of three times independent experiments. ****, $P < 0.0001$, unpaired Student's $t$ test. (B) IFA of tachyzoites in PV with or without ATc and the apicoplast was stained with rabbit anti-CPN60. (C) IFA of extracellular parasites was pretreated with or without ATc, and the apicoplast was stained with rabbit anti-CPN60. Scale bar 5 $\mu$m. (D) Western blot of CPN60 with or without CaM. (E) Diagram and IFA of TgMyoF C-terminal tagged by 3HA in the iCaM strain. (F) IFA of TgMyoF-3HA in the presence or absence of CaM. Scale bar 5 $\mu$m.

derived from a red alga's secondary endosymbiosis (29, 30). Many critical metabolic processes occur in the apicoplast, and parasites lacking the apicoplast exhibit the delayed death phenotype (death in the next lytic cycle) (31, 32). After 24 h of ATc, most tachyzoites had lost their apicoplast, as revealed by staining for the apicoplast marker CPN60 (Fig. 5A and B). Notably, some parasites lacking CaM had apicoplasts stained with anti-CPN60, but these apicoplasts were mispositioned in the parasite's posterior (Fig. 5C). Western blot analysis showed the accumulation of the CPN60 precursor and reduction of the mature form upon CaM depletion (Fig. 5D). These findings indicated that CaM

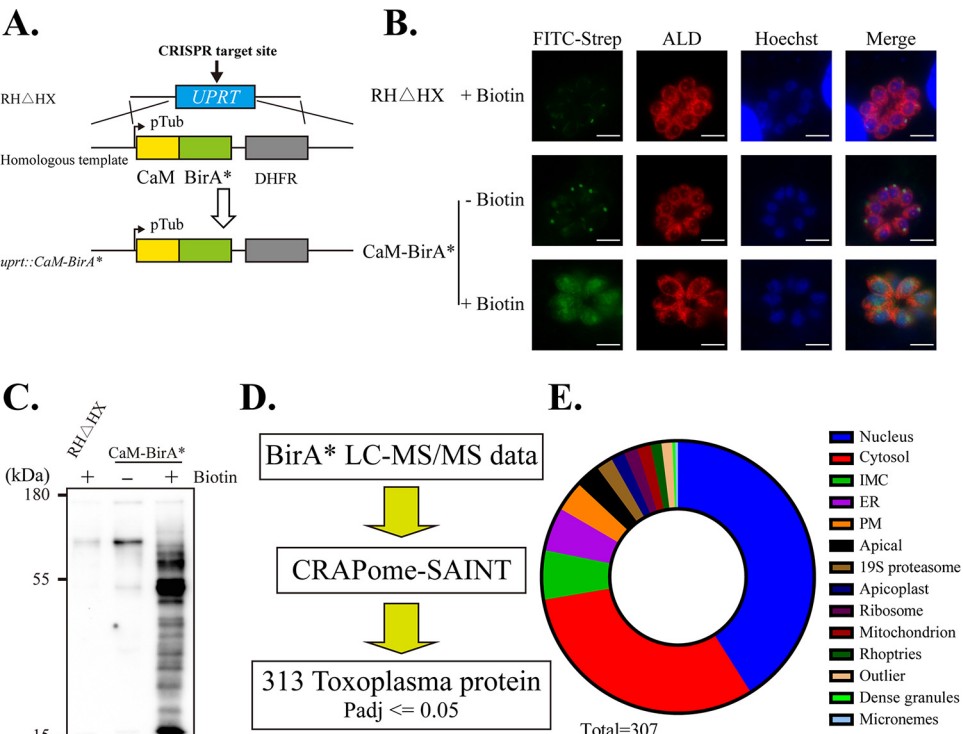

**FIG 6** Identification of CaM-interacting proteins through BioID technology *in vivo*. (A) Construction of CaM-BirA* strain through expressing a CaM-BirA* in the *uprt* locus. (B and C) IFA and Western blot to identify CaM-BirA* strain. Scale bar 5 $\mu$m. FITC-Streptavidin (B) and HRP-Streptavidin (C) were used to stain the CaM proximal interacting proteins labeled with biotin. (D) Route diagram of processing CaM-BioID data. (E) Subcellular localization of CaM-interacting proteins obtained from (D).

depletion-induced defects resulted in apicoplast loss. This apicoplast loss could be the primary reason for the delayed death phenotype of parasites without CaM.

Myosin F (TgMyoF), one of the myosin motors, is mainly found in the vicinity of the apicoplast during tachyzoite division, and it is required for centrosome positioning and apicoplast inheritance (33). In our subsequent *in vivo* BioID experiments, TgMyoF was identified as a CaM proximal interacting protein. TgMyoF has six IQ motifs in its neck domain, and it could interact with myosin light chains (MLCs), such as CaM and CaM-like proteins. Like CaM depletion, TgMyoF depletion results in apicoplast loss in *T. gondii* (33). Thus, the C terminus of TgMyoF was tagged with 3HA in the iCaM strain. IFA results indicated that CaM was colocalized with TgMyoF in the base of tachyzoites in PV and the vicinity of the apicoplast and that CaM depletion changed TgMyoF's location (Fig. 5E and F).

**CaM interacted with over 300 proteins in tachyzoites.** CaM's function is determined by its interacting proteins. In combination with high-resolution liquid chromatography-tandem mass spectrometry (LC-MS/MS), BioID technology can effectively identify proximal interacting proteins in live mammalian cells and parasites (such as *T. gondii*) (34–36). In our study, CaM was fused with BirA*, a promiscuous biotin ligase mutant (R118G), and inserted into the *uprt* locus to biotinylate interacting proteins of CaM (Fig. 6A). IFA and Western blot analyses demonstrated that BioID technology effectively identified CaM proximal interacting proteins (Fig. 6B and C). This study aimed to identify candidate proteins interacting with CaM in *T. gondii* during the tachyzoite stage. First, candidate *T. gondii* proteins were filtered using the significance analysis of INTeractome (SAINT) algorithm, which was designed to identify interacting partners based on affinity purification mass spectrometry data (Fig. 6D) (37, 38). Second, these candidate proteins were ranked by the length-normalized spectral counts (NormSpC) (39, 40). The related results were presented in Table S1 in Supplemental File 1.

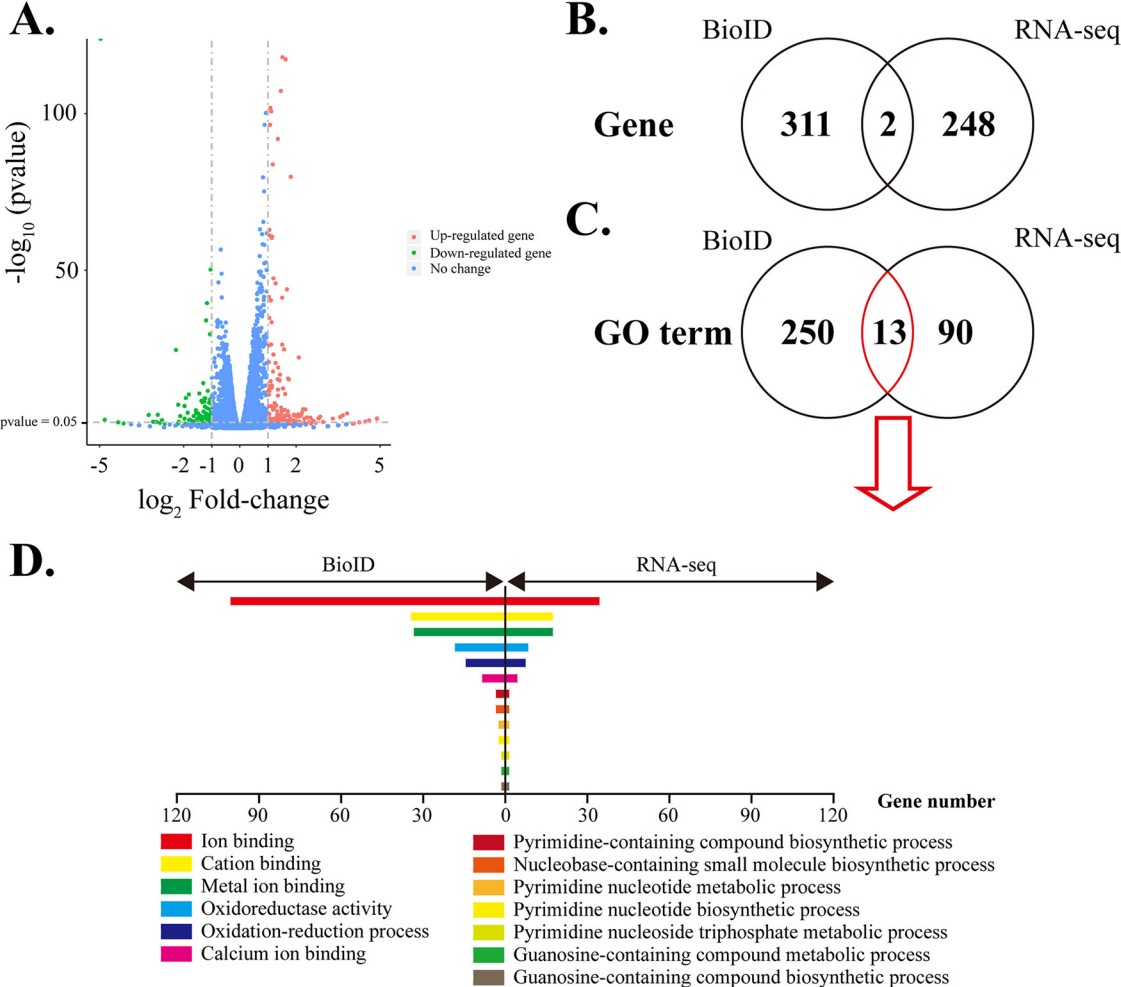

**FIG 7** The transcriptome of iCaM pretreated with or without ATc. (A) Volcano plot of differentially expressed genes upon CaM depletion. (B) Veen of the genes from the differentially expressed genes of transcriptome and the CaM interacting protein-associated genes in BioID data. (C) Venn of GO-terms enriched by the differentially expressed genes of transcriptome and the CaM interacting protein-associated genes in BioID data. (D) Column graph of GO terms from (C).

Our CaM-BioID results demonstrated that 313 *T. gondii* unique proximal interacting proteins had the potential to interact with CaM in the tachyzoite stage. These partners were widely distributed in the whole parasite (such as the cytosol, nucleus, and IMC). These partners' subcellular localization data were derived from the hyperplexed localization of organelle proteins by isotope tagging (hyperLOPIT) proteomics data set (Fig. 6E) (41). The fitness scores of 80% of CaM partners were under 0, and those of 60% of CaM partners were under −2, indicating their potential essential roles in the parasite lytic cycle (Table S1 in Supplemental File 1) (42). These identified CaM partners were subjected to GO (gene ontology) analysis (Table S2 in Supplemental File 2).

**Transcriptome revealed functions of CaM in the tachyzoite stage.** To explore the functions of CaM in *T. gondii*, the transcriptome samples were collected with the treatment of ATc or ethanol for 36 h. Our RNA-seq showed that 262 differentially expressed genes in *T. gondii* (171 upregulated and 91 downregulated genes) were identified after CaM depletion (Fig. 7A, Table S3 in Supplemental File 3). Further, GO analysis of these upregulated and downregulated genes were performed (Table S4 in Supplemental File 4). Only two differentially expressed genes were shared by both the BioID and RNA-seq data. The BioID and RNA-seq data showed that the *T. gondii* differentially expressed genes were enriched in 13 GO terms, such as ion binding, cation binding, metal ion binding, oxidoreductase activity, calcium ion binding, and oxidation-reduction

processes (Fig. 7B to D), indicating that calmodulin was essential for $Ca^{2+}$-involved biological processes.

## DISCUSSION

*Toxoplasma gondii* is an active parasite in many biological processes, such as egress and invasion. Many of *T. gondii*'s biological processes are dependent on $Ca^{2+}$ signaling. Calmodulin is highly conserved as a ubiquitous $Ca^{2+}$ receptor protein and mediates numerous $Ca^{2+}$-dependent events in eukaryotes. It can interact with more than 300 target proteins in humans (43–45). In *T. gondii*, the functions and targets of CaM are rarely known due to its complicated life cycle and various host cells (such as fibroblasts, neurons, and immune cells). The functions of CaM were mainly identified through its antagonists (such as trifluoperazine and calmidazolium) (46–48). Few CaM-regulated partners had been characterized in *T. gondii* tachyzoites. Our results showed that CaM could be efficiently knocked down when the 7tetO-CaM minimal endogenous promoter element was inserted at the N terminus in the TATI strain (Fig. 1A to D). This result suggests that the tetracycline-induced endogenous CaM promoter might be more efficient for CaM transcription than the SAG1 mini promoter in the tetracycline-off system in *T. gondii*. Furthermore, the tetracycline-off system with the target gene's endogenous promoter exhibited great potential to improve the efficiency and compatibility for characterizing essential genes in *T. gondii* and other organisms.

CaM in *T. gondii* tachyzoites has been reported to be localized to the apical and basal end through the fluorescent immunolabeling test with mouse monoclonal anti-calmodulin antibody (*Dictyostelium*, bovine, rat, and chicken) and endogenous CaM fused with YFP, respectively (23, 24). In our study, CaM was localized to the base, cytoplasm, and apex of *T. gondii* tachyzoites. The localization was verified using the N-terminal ty-tag in the iCaM strain (Fig. 1D). The different locations (base, cytoplasm, and apex) of CaM indicated its various functions. For example, CaM at the PV's basal end of tachyzoites indicated its role in the parasite basal pole. The parasites without CaM exhibited abnormal basal constriction, disorderly intravacuolar arrangement, and opposite division orientation of daughter tachyzoite in PV (Fig. 3 and Fig. S3 in Supplemental File 6). Some CaMs were distributed in the vicinity of the apicoplast (present in nearly all members of the Apicomplexa phylum) (Fig. 5E). Upon CaM depletion, the apicoplast loss phenotype also indicated the relationship between CaM and apicoplast (Fig. 5A to D). The apicoplast is mainly involved in the synthesis of isoprenoids, fatty acids, heme, lipoic acid, and iron-sulfur cluster, and loss or dysfunction of the apicoplast can block the parasite's metabolic pathway (49–51). The accumulation of amylopectin reflected parasites' metabolic defects after CaM depletion (Fig. S4 in Supplemental File 6). Our study showed that CaM was partially colocalized with TgMyoF, mainly located in the vicinity of the apicoplast and the basal end of parasites (Fig. 5E) (33). In addition, CaM depletion also affected the localization of TgMyoF in tachyzoites (Fig. 5F). Due to the possible interaction between TgMyoF and CaM based on CaM-BioID data and the Co-IP experiment, we speculated that CaM depletion might affect the function of TgMyoF, which has been reported to be essential for the apicoplast inheritance (Fig. S5A and B in Supplemental File 6) (33). In this study, apicoplast loss might be responsible for the delayed death phenotype of CaM-depleted parasites. CaM and TgMyoF are resistant to knockout, so it is challenging to verify how CaM participates in apicoplast inheritance through MyoF.

CaM is indispensable in the tachyzoite stage, and CaM depletion will destroy replication, invasion, and egress (Fig. 2). Although CaM was distributed in the apical, CaM depletion did not affect the conoid protrusion induced by A23187 (Fig. 4A and B). CaM-like proteins localized at the conoid (such as CaM-like proteins 1, 2, and 3) may have compensated for some of CaM's function. However, the microneme secretion defect induced by CaM depletion was not compensated by raising the cytoplasmic $Ca^{2+}$ with A23187 (Fig. 4C). The CaM depletion-induced phenotypes confirmed that the function of CaM in *T. gondii* was too complicated to be explained from one perspective. To better understand the role of CaM in *T. gondii*, we captured its interacting

partners in the tachyzoite stage through proximity biotinylation technology. Our study identified over 300 *T. gondii* proteins to interact with CaM in the tachyzoite stage. These 300 *T. gondii* partners were widely distributed in various subcellular structures (such as cytosol, nucleus, and IMC) that corresponded to the multiple localization of CaM in tachyzoites (Fig. 6E). Co-IP results proved the interaction between CaM and MyoF, which was identified in the CaM-BioID data (Fig. S5A and B in Supplemental File 6). Although MyoJ is localized at the posterior of the parasite, the CaM-BioID data and Co-IP result indicated that MyoJ did not interact with CaM. (Fig. S5A and C in Supplemental File 6). These Co-IP results demonstrated the reliability of CaM-BioID data. However, the relatively long biotinylation time might cause some underlying problems. For example, some irrelevant proteins were labeled, and some transient critical interactions were neglected. Future research should use faster TurboID enzymes and other protein interaction technologies to shorten reaction times and improve result reliability (52). Future immunoprecipitation experiments might be needed to identify potential CaM-binding proteins and to confirm our CaM-BioID data even more.

In addition, further optimization of the BioID method will provide more detailed information about the proteins interacting with CaM under different stages (such as glide, invasion, and egress). For example, the optimized BioID method can be used to evaluate differences in the interaction partners among different *Toxoplasma* strains or even among the closely related apicomplexan parasite plasmodium under different conditions (such as alkaline treatment). Furthermore, combining the BioID method with other complementary approaches will provide more insight into the CaM interactome. Although the biotin proximity-labeling approach we used in this study is powerful, it has some limitations. First, the CaM C-terminal labeling with BirA* peptide (35.1 kDa) may change CaM's initial structure and affect the natural interaction of CaM. Therefore, it is necessary to verify the integrity and accuracy of interaction partners through immunoprecipitation experiments or partner's biotin proximity-labeling approach in the future.

Our results revealed that CaM was indispensable for the *Toxoplasma* tachyzoite stage. First, CaM may participate in apicoplast inheritance through its interaction with MyoF. Second, CaM may play an essential role in the tachyzoite basal constriction and affect its rosette-like arrangement in PV. In addition, the CaM interactome was identified by biotin proximity-labeling coupled with LC-MS/MS. Our findings provide an insight into CaM function in *Toxoplasma gondii* and other organisms.

## MATERIALS AND METHODS

**Parasite strains.** The type I *T. gondii* strains TATI and RHΔhxgprt were used as the parents of the transgenic strains in this study. All the *T. gondii* strains were maintained in human foreskin fibroblasts (HFFs) cultured in Dulbecco's modified Eagle's medium (DMEM) supplemented with 2% fetal bovine serum, 4 mM glutamine, and 1% penicillin-streptomycin (D2), as previously described (53).

**Plasmids and parasite strain construction.** The plasmid pCaM::DHFR* used for direct knockout of CaM with the DHFR* was constructed with the ClonExpress MultiS One Step Cloning kit (Vazyme, China) by cloning DHFR* and CaM 5H and 3H homology arms into the pUC19 vector. The plasmid pDHFR*-TetO-SAG1-ty-CaM used for CaM knockdown was constructed by the following steps. Firstly, the intermediate plasmid pTetO-SAG1-tyCaM-3H was generated by cloning the CaM coding sequence (CDS) and its 3H homology arm into the linearized vector p7TetOS1 (a gift from the L. David Sibley lab at the Washington University in St. Louis). Subsequently, DHFR* and CaM 5H homology arms were cloned to the KpnI restriction enzyme site by the methods described above. The plasmid pDHFR*-TetO-CaM promoter-ty-CaM was constructed based on the plasmid pDHFR*-TetO-SAG1-ty-CaM using the methods described above. The plasmid pUPRT::CaM-BirA* expressing CaM-BirA* at the *UPRT* locus was constructed by replacing the GRA1 sequence of the plasmid pUPRT::GRA1-BirA* with CaM CDS using the above-mentioned cloning kit (36).

The CRISPR plasmid and the homologous template were cotransfected into the parent strain TATI or RHΔhxgprt at a molar ratio of 5:1. The *T. gondii* stable transfectants were selected with 1 $\mu$M pyrimethamine or 20 $\mu$M 5′-fluo-2′deoxyuridine (FUDR). The single clone was obtained through limiting dilution and diagnostic PCRs.

All homologous recombination and electro-transformation fragments were amplified through Phanta Max Super-Fidelity DNA polymerase (Vazyme, China). All CRISPR-Cas9 target plasmids were constructed by site-directed mutagenesis of the *UPRT*-targeting sgRNA in the original CRISPR/Cas9 template plasmid (pSAG1-Cas9-sgUPRT) with gene-specific sgRNA using the Q5 site-directed mutagenesis kit (New England BioLabs, USA), as described previously (54, 55). All the primers used in this study were listed in Table S5 in Supplemental File 5.

**Immunofluorescence assay.** Intracellular parasites were fixed with 4% paraformaldehyde (PFA, Sigma-Aldrich, USA) in PBS for 15 min, permeabilized with 0.1% Triton X-100 (Sigma-Aldrich, USA) in PBS for 15 min, and blocked with 10% fetal bovine serum in PBS (blocking buffer) for 15 min at room temperature. The samples were incubated with primary antibody and then secondary antibody dilution buffer at 37°C for 30 min. The 6 primary antibodies used in our study included mouse anti-Ty monoclonal antibody (a kind gift from the L. David Sibley lab at Washington University School of Medicine), mouse/rabbit anti-HA (Medical & Biological Laboratories CO., LTD., Japan), rabbit anti-ALD (a kind gift from the L. David Sibley lab at Washington University School of Medicine), rabbit anti-CPN60 (a kind gift from Honglin Jia at Harbin Veterinary Research Institute in China), rabbit anti-GRA7 (stored in our laboratory), and rabbit anti-IMC1 (stored in our laboratory). The secondary antibodies used in our study included Alexa Fluor 488 goat anti-mouse or rabbit IgG(H+L) cross-adsorbed secondary antibody and Alexa Fluor 594 goat anti-mouse or rabbit IgG(H+L) cross-adsorbed secondary antibody (Invitrogen, USA). Cell nuclei were stained with Hoechst 33342 (Beyotime, China). Fluorescent images were captured using an Olympus BX53 microscope (Olympus, Japan) equipped with a ZEISS Axiocam 503 mono camera (Carl Zeiss, Germany).

**Western blot.** Freshly egressed tachyzoites were collected by manual scraping, filtered through a 3.0 $\mu$m-pore size nucleopore hydrophilic membrane (Whatman, USA), washed with PBS, and lysed in 1× SDS-loading buffer (50 mM Tris-Cl, 2% SDS, 0.1% bromophenol blue, 10% glycerin, 20 mM dithiothreitol). Lysates were boiled for 10 min in 1× SDS-loading buffer, separated on 10% polyacrylamide gels by SDS-PAGE, and transferred to PVDF membranes. The membranes were blocked with 1% BSA in TBS and then incubated with primary antibodies diluted in a blocking buffer containing 0.1% Tween 20. The membranes were washed five times with TBS containing 0.1% Tween 20, then incubated with HRP-labeled goat anti-rabbit secondary antibodies or anti-mouse secondary antibodies (Beyotime, China), and detected with Clarity ECL Western Blotting Substrates (BIO-RAD, USA). Subsequently, the membranes were washed with TBS containing 0.1% Tween 20 and scanned on a Tanon 5200 imager (Tanon, China).

**Plaque formation assay.** A confluent monolayer of HFF was infected with freshly egressed 200 parasites (tachyzoites) in a 6-well plate and treated with or without ATc for 10 days. Then the samples were fixed with 4% PFA in PBS and stained for 15 min at room temperature with 0.1% crystal violet (Solarbio, China). Plaque sizes were measured with Adobe Photoshop.

**Intracellular replication assay.** The iCaM and TATI parasites were pretreated with or without ATc for 48 h. Fresh parasites were collected by mechanically disrupting host cells and allowed to grow for 24 h on a coverslip, followed by fixation with 4% PFA in PBS. IFA based on rabbit anti-ALD antibody and mouse anti-SAG1 antibody was performed to distinguish intracellular from extracellular parasites. The number of parasites per vacuole was counted, and the parasites in about 100 vacuoles under each condition were calculated.

**Invasion and egress.** To perform the invasion assay, iCaM and TATI parasites were pretreated with or without ATc for 48 h. Fresh parasites were harvested by mechanically disrupting host cells, resuspended in the D2 medium, and transferred to HFF monolayer coverslips in 24-well plates (1 × 10^6 tachyzoites/well) and incubated with or without ATc for 20 min at 37°C. The invasion process was stopped by adding 4% PFA. Extracellular parasites were stained with mouse anti-SAG1 and then washed gently with PBS. The intracellular parasites in the host cell were permeabilized with 0.1% Triton X-100 and stained with rabbit anti-ALD. The invasion efficiency was expressed as the ratio of parasite number to host cell number.

To examine the egress of parasites, fresh parasites were harvested by mechanically disrupting host cells, resuspended in the D2 medium, and added to HFF monolayer coverslips in 24-well plates incubated with or without ATc for 36 h. After 36 h of incubation, the medium was replaced with 2 $\mu$M A23187 or DMSO, followed by a 15-minute incubation (Sigma-Aldrich, USA). The egress process was stopped by adding 4% PFA. The samples were first stained with primary antibodies, mouse anti-SAG1 (whole parasite), and rabbit anti-GRA7 (parasite vacuole), then stained with the secondary antibodies Alexa Fluor 488 goat anti-mouse and Alexa Fluor 594 goat anti-rabbit. At least 100 vacuoles were observed per treatment in three independent biological experiments. Only vacuoles containing over two parasites were counted in every coverslip.

**Basal features of tachyzoite and conoid extrusion.** The extracellular parasites were collected by centrifugation, and then the intracellular parasites were collected after repeated pipetting several times with a syringe. The basal feature of tachyzoites in the CaM + ATc and control was observed and assessed, respectively. The conoid protrusion of parasites was stimulated by A23187, monitored, and evaluated in the experiment and control group.

**Daughter tachyzoite division orientation.** Fresh parasites were collected by mechanically disrupting host cells and allowed to grow for 24 h on a coverslip, followed by fixation with 4% PFA. Daughter tachyzoites were visualized by IFA using rabbit anti-IMC1 antibody, and their division orientation was observed and counted. The data from three independent biological replicates were expressed as means ± SD, and the division orientations of 100 parasites were observed under each condition.

**Apicoplast loss.** The iCaM and TATI parasites were pretreated with or without ATc for 48 h. Fresh parasites were collected by mechanically disrupting host cells and stuck onto a coverslip coated with 1 mg/mL poly-L-lysine (Beyotime, China), followed by fixation with 4% PFA. IFA was performed using rabbit anti-CPN60 and mouse anti-SAG1 antibodies. The data from three independent biological replicates were expressed as means ± SD; the apicoplast number in 100 parasites was counted under each condition.

**BioID sample collection.** The RHΔhxgprt-CaM-BirA* strain was used to infect HFF cells for about 24 h and then incubated 24 h with or without 150 $\mu$M biotin (Sigma-Aldrich, USA) in the D2 medium, as

previously described (56). The parasite samples were collected by manual scraping, washed in PBS, and lysed in RIPA buffer (50 mM Tris, pH 7.4; 150 mM NaCl; 1% Triton X-100; 1% sodium deoxycholate; and 0.1% sodium dodecyl sulfate [SDS]; Beyotime, China) supplemented with protease inhibitor (ThermoFisher, USA) (36). The lysates were centrifuged at 12,000 × $g$ for 10 min at 4°C. The supernatants were incubated for 6 h at 4°C with gentle tube rotation with MyOne Streptavidin magnetic beads T1 (ThermoFisher, USA). Magnetic beads were washed three times with RIPA buffer, then boiled for 10 min in 1× SDS-loading buffer to release biotinylated proteins. 10% of each sample was analyzed by Western blotting using HRP-labeled streptavidin (Beyotime, China), while the remaining samples were loaded in 12% SDS-PAGE gel and separated for about 10 min. The top gel containing biotinylated proteins was cut, freeze-dried, and used for LC-MS/MS.

**Mass spectrometry analysis and data processing.** The gel band samples were reduced, alkylated, and then washed with ammonium bicarbonate/acetonitrile to remove stains and SDS. Trypsin was added, and digestion was carried out overnight at 37°C. Peptides were extracted from the gel pieces, dried, and redissolved in 2.5% acetonitrile and 0.1% formic acid. Each digest was subjected to nanoLC-MS/MS using a 2-h gradient on a 0.075 mm × 250 mm $C_{18}$ column feeding into a Q-Exactive HF mass spectrometer.

All MS/MS samples were analyzed using Mascot (Matrix Science, London, UK; version 2.6.1). The Mascot was set to search the cRAP_20150130.fasta and Tgondii_GT1_20170914 database (8,462 entries), assuming the digestion enzyme trypsin. The Mascot was set with a fragment ion mass tolerance of 0.060 Da and a parent ion tolerance of 10.0 PPM. Deamidated asparagine and glutamine, oxidation of methionine, and carbamidomethyl of cysteine were specified in Mascot as variable modifications.

Scaffold (version 4.8.4, Proteome Software Inc., Portland, OR) was used to validate MS/MS-based peptide and protein identifications. Peptide identifications were accepted if they could be established at >80.0% probability by the Peptide Prophet algorithm with Scaffold delta-mass correction (57). Protein identifications were accepted if they could be established at >99.0% probability, and they contained at least two identified peptides. Protein probabilities were assigned by the Protein Prophet algorithm (58). Proteins that contained similar peptides and could not be differentiated based on MS/MS analysis alone were grouped to satisfy the principles of parsimony. Proteins sharing significant peptide evidence were grouped into clusters. Spectral counts of *T. gondii* proteins were used to analyze the probability of interaction between CaM and its prey using the CRAPome repository with SAINTexpress (38, 59). Normalized spectral counts (NormSpC) were calculated, as previously reported (39, 40) using the following formula: $NormSpc_{i,j} = length(aa)/(length[aa]_j) \times (AvgSpeC_{i,j} - AvgSpe_{control\ i,j})$. For bait $i$ (CaM) and prey $j$, the average spectral counts for $j$ detected within bait $i$ ("biotin" group) were first subtracted from those for $j$ detected within bait $i$ ("no biotin" control group). The value obtained in the previous step was then multiplied by the median length of all prey detected in bait $i$, and divided by the length of $j$.

**Co-Immunoprecipitation.** The iCaM-MyoF-HA and iCaM-MyoJ-HA strains were constructed through a C-terminal tagged with 3HA in the iCaM parent strain as described above. The fresh parasites of iCaM-MyoF-HA, iCaM-MyoJ-HA, and iCaM control were all collected and lysed in RIPA buffer, respectively. The sample supernatant was incubated with mouse anti-HA and then protein A+G magnetic beads (Beyotime, China). The magnetic beads were boiled in a 1× SDS-loading buffer to release prey proteins. These samples were separated on SDS-PAGE and then used for Western blotting.

**Transcriptome.** iCaM parasites were pretreated with or without ATc for about 40 h. Fresh parasites were harvested by mechanically disrupting host cells and washed with PBS. Total RNA was extracted using Transzol UP reagent (TransGen Biotech, China), followed by qualification using the Qubit RNA assay kit in the Qubit 2.0 Fluorometer (Life Technologies, USA). Total RNA integrity was detected using the Bioanalyzer 2100 system (Agilent Technologies, USA). The RNA library was constructed using the NEBNext Ultra RNA Library Prep kit for Illumina (New England BioLabs, USA), following the manufacturer's instructions. The mRNA was purified and used for cDNA synthesis and Illumina Hiseq platform sequencing. The clean data were obtained by removing low-quality reads and those containing adapters from the raw data. The clean data were aligned to the *Toxoplasma gondii* TgGT1 genome (ToxoDB-43) using HISAT2 v2.0.5. Differential expression analysis was performed using the DESeq2 R package (1.16.1) with a default parameter, and GO analyses were performed by the clusterProfiler R package (4.2.0) with the default parameter (60). The GO terms with a corrected $P$ value less than 0.05 were considered significant.

**Data availability.** The transcriptome data generated for this study can be found in the Gene Expression Omnibus (GEO), accession no. GSE198001. The mass spectrometry proteomics data for CaM-BioID were deposited to the ProteomeXchange Consortium via the PRoteomics IDEntifications (PRIDE) partner repository with the data set identifier PXD032102.

## SUPPLEMENTAL MATERIAL

Supplemental material is available online only.
**SUPPLEMENTAL FILE 1**, XLSX file, 0.1 MB.
**SUPPLEMENTAL FILE 2**, XLSX file, 0.04 MB.
**SUPPLEMENTAL FILE 3**, XLSX file, 0.1 MB.
**SUPPLEMENTAL FILE 4**, XLSX file, 0.02 MB.
**SUPPLEMENTAL FILE 5**, XLSX file, 0.01 MB.
**SUPPLEMENTAL FILE 6**, PDF file, 1.2 MB.

## ACKNOWLEDGMENTS

We are thankful to Sophie Alvarez and Michael Naldrett from the Proteomics and Metabolomics Facility (PMF team), Center for Biotechnology at the University of Nebraska-Lincoln, for their kind help on high-quality mass spectrometry services and the facility and instrumentation supported by the Nebraska Research Initiative. Project support was kindly provided by the National Natural Science Foundation of China (Grant No. 31272553).

We declare no conflict of interest.

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
