## [Reviewer comments · Microbiology Spectrum]

Microbiology Spectrum

Essential functions of calmodulin and identification of its proximal interacting proteins in tachyzoite-stage *Toxoplasma gondii* via BiOLD technology

Yongle Song, Longjiao Li, Xinyu Mo, Ming Pan, Bang Shen, Rui Fang, Min Hu, Junlong Zhao, and Yanqin Zhou

Corresponding Author(s): Yanqin Zhou, Huazhong Agricultural University

Review Timeline:

Submission Date:	April 15, 2022
Editorial Decision:	July 25, 2022
Revision Received:	August 16, 2022
Accepted:	August 31, 2022

Editor: Björn Kafsack

Reviewer(s): The reviewers have opted to remain anonymous.

Transaction Report:

DOI: <https://doi.org/10.1128/spectrum.01363-22>

July 25, 2022

Prof. Yanqin Zhou
Huazhong Agricultural University
Wuhan
China

Re: Spectrum01363-22 (Essential functions of calmodulin and identification of its proximal interacting proteins in tachyzoite-stage *Toxoplasma gondii* via BioID technology)

Dear Prof. Yanqin Zhou:

I want to apologize for the length of the review process but unfortunately securing more than one reviewer was a challenge for this manuscript.

I therefore decided to evaluate the manuscript myself. Indeed, I agree with the other reviewer that the presented work is well executed and controlled.

Your findings will be a valuable addition to the literature.

However, the English usage and grammar in the manuscript are not to the standards required for ASM Spectrum Microbiology. I have made a few edits myself in the attached PDF but I strongly suggest using a language editing service with biomedical expertise. Several options can be found here: <https://journals.asm.org/language-editing-services>

Thank you for submitting your manuscript to Microbiology Spectrum. As you will see your paper is very close to acceptance. Please modify the manuscript along the lines I have recommended. As these revisions are quite minor, I expect that you should be able to turn in the revised paper in less than 30 days, if not sooner. If your manuscript was reviewed, you will find the reviewers' comments below.

When submitting the revised version of your paper, please provide (1) point-by-point responses to the issues I raised in your cover letter, and (2) a PDF file that indicates the changes from the original submission (by highlighting or underlining the changes) as file type "Marked Up Manuscript - For Review Only". Please use this link to submit your revised manuscript. Detailed instructions on submitting your revised paper are below.

Link Not Available

Sincerely,

Björn Kafsack

Reviewer comments:

Reviewer #1 (Comments for the Author):

- It is odd that the authors chose to use tet-off instead of AID, especially given previous success with the AID system and other CaM-like proteins in *Toxoplasma*. The authors should point out that the timing (days) of treatment make it impossible to discern direct from indirect phenotypes.

- where possible, phenotypes should be quantified with number of parasites and error noted rather than using inexact language, e.g. line 163 "slightly larger posterior ring". Line 166 "great effect on basal complex constriction". What does "great effect" mean?

- it would be helpful (and greatly improve the utility of the data) if the authors examined their BioID hits for CaM-binding motifs - how many of these hits might be potentially direct versus indirect?

- Tet-off is not a knockout approach, indirect or otherwise. It is a transcriptional repression system, typically referred to as "knockdown." The authors should use common scientific terminology instead of making up their own.

- While I don't normally like to harp on English usage, there are major issues with the grammar and language usage throughout the manuscript, which make it hard to interpret the science at times and need to be corrected (in spite of the "language polishing"). This is a problem in both the results as well as in the methods (e.g. BioID sample collection lines 470-472 says that the infected hffs were incubated 24h in ddH₂O+biotin which is unlikely... If that's really what the authors did, then none of these data are interpretable at all).

Preparing Revision Guidelines

- point-by-point responses to the issues I raised in your cover letter
- Upload a compare copy of the manuscript (without figures) as a "Marked-Up Manuscript" file.
- Each figure must be uploaded as a separate file, and any multipanel figures must be assembled into one file.
- Manuscript: A .DOC version of the revised manuscript
- Figures: Editable, high-resolution, individual figure files are required at revision, TIFF or EPS files are preferred

Please return the manuscript within 60 days; if you cannot complete the modification within this time period, please contact me. If you do not wish to modify the manuscript and prefer to submit it to another journal, please notify me of your decision immediately so that the manuscript may be formally withdrawn from consideration by Microbiology Spectrum.

To editor:

Thank you for your valuable review and suggestions about our manuscript. We have tried our best to revise the manuscript according to your kind comments and suggestions.

To reviewer #1 (Comments for the Author):

Q1- It is odd that the authors chose to use tet-off instead of AID, especially given previous success with the AID system and other CaM-like proteins in *Toxoplasma*. The authors should point out that the timing (days) of treatment make it impossible to discern direct from indirect phenotypes.

Answer:

Although the AID system has been successfully applied to the conditional knockout of some essential genes in *Toxoplasma gondii*, there were concerns that the AID element might be too large to affect the function of CaM (16.8 kDa) when the CaM gene knockdown project was carried out. The David Sibley laboratory tried to degrade the CaM (the object of our research) using the AID system, but it was ultimately unsuccessful. So, we decided to use tet-off system. In order to avoid the indirect phenotype caused by CaM deletion as much as possible, we performed Western blot to identify the amount of CaM protein at 0, 24, 36 and 48 hours after adding ATC. The related result was in Fig 1C.

Q2- where possible, phenotypes should be quantified with number of parasites and error noted rather than using inexact language, e.g. line 163 "slightly larger posterior ring". Line 166 "great effect on basal complex constriction". What does "great effect" mean?

Answer:

Thanks for your good suggestion. We have reviewed the manuscript in detail and corrected relevant inaccuracies. The "slightly larger posterior ring" was inexact language, the similar mistakes were changed to "the abnormal posterior pole". The "great effect on basal complex constriction" was also inaccurate, the similar mistakes were changed to "had an effect on basal complex constriction".

Q3- it would be helpful (and greatly improve the utility of the data) if the authors examined their BioID hits for CaM-binding motifs - how many of these hits might be potentially direct versus indirect?

Answer:

Thanks for your good suggestion. We searched the Calmodulin target database (<http://calcium.uhnres.utoronto.ca/ctdb/ctdb/sequence.html>) for possible calmodulin-binding motifs in these BioID hits. These results were added in the table S1.

Q4- Tet-off is not a knockout approach, indirect or otherwise. It is a transcriptional repression system, typically referred to as "knockdown." The authors should use common scientific terminology instead of making up their own.

Answer:

We agree with this suggestion and have modified the terminology throughout the text as appropriate. The “knockout” used here was not suitable for the tet-off system. The modification has been made in the revision as suggested.

Q5- While I don't normally like to harp on English usage, there are major issues with the grammar and language usage throughout the manuscript, which make it hard to interpret the science at times and need to be corrected (in spite of the "language polishing"). This is a problem in both the results as well as in the methods (e.g. BioID sample collection lines 470-472 says that the infected hffs were incubated 24h in ddH₂O+biotin which is unlikely... If that's really what the authors did, then none of these data are interpretable at all).

Answer:

We apologize for the poor language of our manuscript. We have now worked on both language and readability and have also involved native English speakers for language corrections. We really hope that the language level has been substantially improved. BioID sample collection lines 470-472 was not correct. We apologize for the confusion generated by the previous version of the manuscript. We have corrected to “incubated 24 h with or without 150 μM biotin in the D2 medium”. The biotin was dissolved in ddH₂O. *T. gondii* strains were maintained in human foreskin fibroblasts (HFFs) cultured in Dulbecco’s modified Eagle’s medium (DMEM) supplemented with 2% fetal bovine serum, 4 mM glutamine, and 1% penicillin-streptomycin (D2).

August 31, 2022

Prof. Yanqin Zhou
Huazhong Agricultural University
Wuhan
China

Re: Spectrum01363-22R1 (Essential functions of calmodulin and identification of its proximal interacting proteins in tachyzoite-stage *Toxoplasma gondii* via BiolD technology)

Dear Prof. Yanqin Zhou:

Your manuscript has been accepted, and I am forwarding it to the ASM Journals Department for publication. You will be notified when your proofs are ready to be viewed.

Sincerely,

Björn Kafsack
Editor, Microbiology Spectrum

Journals Department
Supplemental file 3: Accept
Supplemental file 5: Accept
Supplemental file 2: Accept
Supplemental file 4: Accept
Supplemental file 1: Accept
Supplemental Material: Accept